# Understanding the Contextual Privacy Comprehension, Attitudes, and Behaviours of Users

Anon*

Anon

## ABSTRACT

To understand and protect their privacy in a world of pervasive data collection, users must understand not only disclosure, but also the transmission and third-party use of their data both in the present and future. Researchers label this phenomenon the contextual integrity of information. While many security and privacy researchers seek to design automated tools to enforce contextual integrity through formally specified policies, this paper explores users' understanding of contextual integrity and their ability to influence it as they learn. We explore whether, by educating participants, participants are more likely to understand and control for the long-term implications of information disclosure. We find that participants do develop an understanding of contextual integrity, but continue to struggle with effective methods to protect the contextual integrity of their data. Together, these results provide guidelines for the design of privacy-preserving technologies with learning and comprehension as guiding principles.

## 1 INTRODUCTION

In an era of 'Big Data' privacy becomes increasingly harder for users to understand. Users not only need to decide what they disclose; they now also need to consider to whom that data is disclosed, how the data will be transmitted, how it will be stored, and for how long it can be used. Furthermore, past data and future data may be linked with currently disclosed data and cross-referenced to reveal additional information about the user. Nissenbaum et al. [21, 22] coin the term *contextual integrity* or *contextual privacy* to capture the concept of protecting data based on what, who, how, and when data will be used.

In this paper, we explore a number of questions revolving around users' understanding of contextual privacy and integrity of data. One thing that is known from past work is that there exists a well-documented mismatch between users attitudes and behaviours [13]; however, in the case of contextual privacy, less is known about user attitudes and knowledge as a precursor to inform behaviours. We, therefore, begin by seeking to discern how users contextualise, visualise, and understand contextual privacy. Many of these questions are tied up in decision making processes that underlie a decision to disclose: for example, is information disclosure based upon a cost-benefit analysis, is it an in-the-moment decision making process by end-users, or are there existing normative behaviours that promote different disclosure decisions [5]? Is the disclosure perceived to be persistent or ephemeral? Then, given this understanding of contextual privacy and information disclosure, we explore whether or not we can more fully inform users regarding their decisions to disclose information.

To explore these questions of user conception of contextual privacy and user decision making, this paper presents two studies. We first contrast motivation, knowledge and behaviours with respect to contextual privacy. We also evaluate one instrument designed to

---

*e-mail: anon@email.com

teach users about contextual privacy, an online video [8] designed to increase user awareness of the risks of pervasive data capture. One take-away from our initial study was a stated desire of users to increase how they preserve the contextual integrity of their data. Given this stated desire, we performed a second study that evaluates how successful users might be at enhancing the contextual integrity of their data. We use the above online video tutorial and evaluate participants in-the-moment and then follow up with participants to measure how well participants retain information, and explore whether increased knowledge of contextual privacy helps participants align their attitudes and their behaviours. We find that users struggle to fully apply their new knowledge to match desired change in behaviours. Given these observations, we discuss the need for on-going, in-the-moment guidance for users that identifies the risks to contextual privacy with respect to use of social media, online email accounts, and web search engines.

## 2 RELATED WORK

Contextual privacy falls under the larger definition of *privacy*, which refers the user's right to determine access to their own information [4]. Privacy differs from security, the practices of protecting sensitive data from unwelcome, unauthorized user groups [4, 18, 29, 30]. However, privacy and security are linked [5] in that security is a set of technical concerns whereby privacy, a social concern, is protected.

With the growing complexity and pervasiveness of online data collection and storage, it becomes easy for companies to keep an on-going profile of individuals. Many companies can also cross-reference these sources through users' internet use and personal devices [17], accumulating this information over time to develop an ever-clearer picture of individual users. Arguably, one significant problem is that the data that is collected by companies that provide services to users is not strictly used to benefit the user [15]; instead, these companies often use this data for profit. For instance, data may be sold as a commodity to data brokers or companies that profit from interactions with advertisements. As a result, the consumer or user of a particular product (e.g. social media, search engines, etc.) can actually become the product that is consumed. Despite the plethora of information captured and the existence of privacy laws and privacy expectations, the use of people's personal data on the internet is still unregulated [15].

The ideas presented above – of privacy disclosure not only impacting users in-the-moment and with the recipient but, instead, through transmission and archiving, with others and in the long term – has given rise to the premise of contextual integrity of data or, more generally, contextual privacy [2, 21, 22]. Here we define *contextual privacy* as information integrity beyond it's initial use or context of collection. To promote an understanding of contextual privacy within end-users, we must both understand users and, in particular, the heterogeneity of users. Second, we must also understand how users currently make privacy decisions in the context of interactions with third parties. The remainder of this section addresses these two factors.

### 2.1 Understanding Users

Users are heterogeneous, and it has been further suggested that security and privacy related behaviours are a result of personality or eval-

uation of user types. Classic research in the area by Westin [29, 30] discusses the classification of users based on users' perceptions of privacy and trust. Using a short questionnaire, Westin identified three categories of classification: Privacy Fundamentalists, Privacy Pragmatists and Privacy Unconcerned. Privacy fundamentalists are defined as, "generally distrustful of organisations that ask for their personal information, worried about the accuracy of computerised information and additional uses of it, and are in favour of new laws and regulatory actions to spell out privacy rights and provide enforceable remedies" [29, 30]. Essentially, fundamentalists will choose privacy controls over convenience. The pragmatic are defined as those who "weigh the benefits of consumer opportunities and services, protections of public safety or enforcement of personal morality against the degree of intrusiveness of personal information sought and the increase in government power involved" [29, 30]. Individuals falling under the pragmatic category will reason whether it is worth the privacy risk for convenience, so when benefits outweigh concerns, they will opt into providing information. They also believe they should have this choice. Lastly, the unconcerned are defined as "generally trustful of organisations collecting their personal information, comfortable with existing organisational procedures and uses, are ready to forego privacy claims to secure consumer-service benefits or public-order values, and are not in favour of the enactment of new privacy laws or regulations" [29, 30]. The unconcerned will always choose convenience over privacy.

Westin's categories are commonly used in research (see [6, 11]), and in this work we, also, use Westin's Privacy Segmentation Index. However, there are well-documented challenges to Westin's user categories [6, 11]. First of all, Westin's categories take a uni-dimensional approach toward privacy, specifically focusing on issues of third party trust. To further explore user attitudes, Naresh et al. introduce the Internet Users' Information Privacy Concerns (IUIPC), a more comprehensive representation of online consumers' concerns for information privacy [19]. IUIPC is a ten-item scale that provides an index of privacy concern based on three dimensions: control, awareness (of privacy practices) and collection. From the perspective of the user, control is whether the user has control over the data, awareness is whether the user is adequately informed about the use of data, and collection is whether the exchange of personal information is equitable or fair [19]. To classify users, users complete a questionnaire that includes a series of Likert style questions on behaviours, attitudes, and responses to scenarios.

Alongside the uni-dimensional view of privacy, Westin's categories have been shown to be problematic when attempting to correlate with behaviour. This is because a user's privacy concern is an individual's tendency to worry about privacy *in general*, ignoring contextual cues, and overall is not predictive of disclosure intention or behaviour [23]. The mismatch reported here is called the *Privacy Paradox*, i.e. people tend to say they have higher levels of privacy concern than what their behaviour indicates [23]. In their work on privacy, Dourish and Anderson [5] note that information practices by end users – specifically the decision making about whether to disclose information – may be modelled in three different ways: as economic rationality, as practical action, or as discursive practice. The question revolves around whether or not, during disclosure, users measure the cost versus benefit, whether they just make the decision that "feels right", or whether there are some established norms that guide decision making. To attempt to address the privacy paradox, Dupree et al. [6], cluster users based upon behaviour and then more deeply explore these clusters to understand user categories. Their results preserve two of Westin's categories – fundamentalists and the unconcerned – but break down pragmatics into three additional categories, lazy experts, technicians, and struggling users. They hypothesise that one reason attitudes may correlate poorly with behaviours is because attitudinal scales do not take into account background knowledge.

Despite the challenges with Westin's categories, one significant advantage is that categorisation is quick (three questions) [19] and can be done a priori (does not depend on specific behavioural analysis) [6]. Furthermore, while Westin's categories may correlate poorly with behaviours, they do elicit concerns about and trust in information and about use and disclosure in relation to third parties. As a result, they are a useful insight into users' a priori attitudes toward information disclosure.

## 2.2 Fostering Privacy-Preserving Behaviours

One challenge with the privacy paradox is how we can better align users' intentions and behaviours. In a paper titled: *"Your location has been shared 5398 times!"* [3], researchers evaluated the benefits of giving users an application permission manager. The manager was designed to send 'nudges' intended to raise user awareness of the data collected by the applications currently on their device. After as little as a week of use, over 50% of participants further restricted app permissions based on this feedback. Their results also confirmed that users are generally unaware of mobile application data collection practices.

The above research represents an application of persuasive technology [7], providing both awareness and a 'spark' to activate improved behaviour. However, location information is an in-the-moment disclosure of information. When extending this to contextual privacy, it becomes less clear how we can enable users to understand contextual privacy and act in a way that preserves contextual integrity of their data to the desired level.

One option for overcoming the privacy paradox is simply providing users with information. There is some evidence to support education as a tool for privacy preservation: Paine et al., in early work, found that people's general privacy attitudes and behaviours would more likely be influenced by public education initiatives than factors such as the re-design of services requesting information [24]. Supporting this point, Klasnja et al. [12] found that when presented with security threats, users indicate that they intend to change behaviour. However, in a replication study of Klasnja et al.'s work that actually incorporates follow-up with users regarding whether they follow-through on intent, Swanson et al. [27] found, again, an instance of the privacy paradox, where users did not follow through. However, again, little of this work explores contextual privacy; it, instead, focuses on discrete actions at a single point in time.

## 2.3 Overview of Studies

In the following sections of this paper, we conduct two studies, an initial attitude-behaviour-knowledge study seeking correlations across attributes of users. In this study, we also provide an intervention and seek to understand whether participants wish to change. Motivated by this study, we perform a second study where we repeat the initial study design, but add a follow-up with participants two weeks later to assess their success. The flowchart depicted in Figure 1 provides an overview of our methodology for probing, first, open questions identified by related work, and, second, verification and follow-up of observations during our second study. If methodological confusion arises, this Figure can serve to ground the reader in our methodologies for Study 1 (S1) and Study 2 (S2).

## 3 MEASURING CONTEXTUAL PRIVACY PRACTICES

Our initial goal in this paper is to explore users' understanding of contextual privacy. While a significant body of research has explored privacy in terms of disclosure of specific pieces of information, contextual privacy has received more limited attention. In particular, we wish to determine whether or not users are aware of on-going data collection, whether or not they currently take steps to protect themselves, and whether or not their attitudes match their behaviours.

To evaluate attitudes and behaviours, we perform an online study on a crowdsourcing platform. The study has two parts, an initial

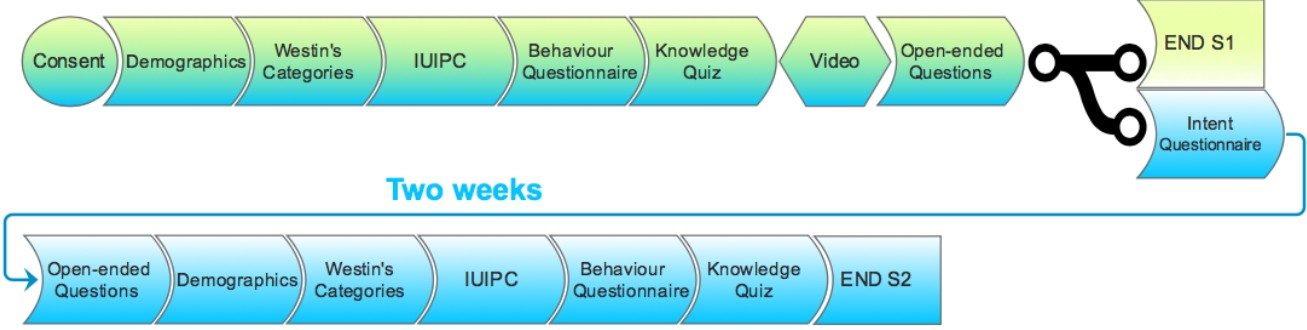

Figure 1: Overview of Protocol for SI & SII

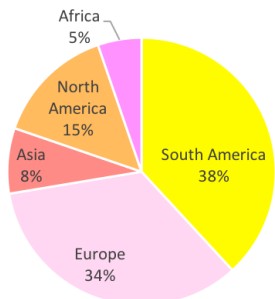

Figure 2: Study I: Participant's current location at the time of the questionnaire

information extraction part, followed by an intervention to educate users and assess whether they acquire new knowledge. Details of this study follow.

### 3.1 Participants

Our survey was conducted on a crowdsourcing platform called CrowdFlower, a platform broadly similar to Mechanical Turk [16]. In total, 76 adult ( > 18 years old) participants completed the study. The average age of participants was 30 years old, and we had 56 male, 19 female, and one participant whom preferred not to specify a gender. Participants were international; Figure 2 illustrates the breakdown by continent.

### 3.2 Protocol

As noted, our study proceeds in two parts: a set of questionnaires designed to elicit participants attitudes and behaviours regarding contextual privacy; and an educational component where we leverage a video [8] created by Reveal, the YouTube channel of the Center for Investigative Reporting, a non-profit that seeks to inform the public on societal issues of import.

#### 3.2.1 Questionnaires

The participants were asked to fill out four questionnaires to help us better understand our user set and to assess potential instruments for our study. The questionnaires had three distinct goals, in order:

- *Demographics*. The first questionnaire captured demographic data including age, gender, and country of origin.

- *Attitudes*. We administered two different standardised questionnaires from the literature that elicit the attitude of the participant towards security and privacy. These were Westin's

Privacy Segmentation [29,30] questions and the Internet Users' Information Privacy Concerns (IUIPC) questionnaire [19].

- *Behaviours*. We investigate current security and privacy practices in the form of their behaviours. We created a questionnaire with a series of scenarios with scaled questions. The scale lists possible actions or choices that provide security to protect sensitive data.

Each of these questionnaires is reproduced in the Appendix, allowing the interested reader to evaluate both the questionnaire and the scoring strategies used. The novel questionnaire is the behaviour questionnaire, which we designed specifically for this study. Its design and scoring are heavily influenced by the IUIPC questionnaire. In particular, we asked a series of 12 questions on ad blocker use, privacy aware browser use/knowledge, downloading practices, search practices including tracking, social media posting and sharing, and the care with which the participant reads terms and conditions notices.

#### 3.2.2 Educational Video Module

After completing the questionnaires, the participants completed an online learning module. The module consists of an interactive quiz administered in CrowdFlower coupled with a YouTube video [8] that explains data collection by online companies.

The video is designed by the Center for Investigative Reporting to explain issues related to contextual privacy, and, in particular, data capture and long-term storage by online service providers. It is designed to have a low technical barrier to comprehension. The video specifically focuses on the use of multiple information sources such as: Facebook, Online Shopping, and Camera tracking. In the video, the main character accomplishes everyday tasks - socialising, shopping, and planning a vacation; simultaneously, the data is visualised as *'leaking'* and being integrated into a larger information profile by companies who would like to better target her in order to increase their profit margins. The interested reader can access the video online [8].

The online learning module is the treatment or intervention which is coupled with the video. The quiz is included in the Appendix as an Online Module Quiz. As the user answers questions, they are given feedback. The quiz is designed to be both an informative learning experience and a method of data collection.

Finally, after the video intervention, we provide a post-video follow-up survey where we assess the video by asking a series of open-ended questions. The questions ask whether the video taught new information and, if so, how participants feel about the information learned (or if not how they feel about the information in the video), and asks participants whether they feel their practices are

currently sufficient, if they intend to make changes, and to elaborate on those points.

## 3.3 Measures

Data was collected and scored as follows:

1. *Demographics*: Demographic data was left in narrative form. We did examine gender, age, and country of origin for correlates with our other data points.

2. *Westin's User Categories*: Westin's categorisation was performed as specified [11, 30]. Raw numbers of participants are reported for each category. Westin's categories were used as categorical variables.

3. *IUIPC*: The IUIPC data was captured and scored in each of the three categories (Control, Awareness, and Collection) as indicated by [19]. Raw scores were used as correlates with values for behaviour and knowledge.

4. *Behaviours*: Our behavioural questionnaire consists of 12 questions, as shown in the Appendix. Similar to Naresh et al.'s IUIPC [19], we assign Likert-based answers ('Does not describe me at all' to 'Completely describes me') to these questions a numerical score in the range [-2, +2] with 0 mapping to neutral or no answer, -2 mapping to practices which decrease privacy, and +2 mapping to practices that increase privacy, similar to IUIPC scoring.

5. *Knowledge Quiz*: The Test Your Knowledge quiz is a set of eight true/false and six multiple choice questions. It is administered before the video and also used during the video as a tool to guide knowledge acquisition.

## 4 STUDY I: RESULTS

In this section, we present Westin's categories as a set of categorical variables of our participants. We first compare Westin's categories to scores for participants attitude (from the IUIPC), behaviour (from our behavioural test) and knowledge (from our quiz scores). Next, we examine the IUIPC results broken down into Control, Awareness, and Collection scores and examine the correlation between numerical IUIPC scores and the scores from our Behaviour questionnaire and our quiz scores.

### 4.0.1 Westin versus IUIPC, Behaviour, and Knowledge

Westin's privacy segmentation is performed using three questions as highlighted in the appendix [13]. We find 11 unconcerned, 53 pragmatists, and 12 fundamentalists in our participant set. We view the three categories as categorical independent variables. Participants were grouped according to their segmentation based upon Westin's categories and then a multi-variate analysis of variance was performed to characterise means. Figure 3 depicts scores for Westin's categories for each of IUIPC, behaviour, and knowledge questionnaire scores.

A Multi-Variate Analysis of Variance (MANOVA) with IUIPC Overall Score, Behaviour Score, and Quiz Score as dependent variables revealed significant differences based on Westin's categorisation of users into groups, $F_{(6,142)} = 2.185, p(0.048) < 0.05, \lambda = 0.838, \eta^2 = 0.085$. Note, however, that, while results are just under significance at the 0.05 level, overall effect size is weak. Post-hoc analysis indicates that only IUIPC score is significant with respect to Westin's categorisation of users. The unconcerned had lower IUIPC scores than pragmatists or fundamentalists (pragmatists versus fundamentalists n.s.).

### 4.0.2 Correlation Analysis

While Westin's categories do segment users, a 3-category view of users lacks fine-grained granularity in terms of directly modelling attitude with respect to behaviours and/or knowledge. To further explore whether attitudinal tests – for example the IUIPC – can be used to predict behaviours and knowledge, we performed a correlation analysis examining the relationship between variables scored from the IUIPC and variables scored from the behaviour and knowledge questionnaires. Table 1 depicts these results. Statistically significant correlations are highlighted.

Note, first, that overall IUIPC score correlates well with control, awareness, and collection scores, an expected result as the IUIPC score is comprised of control, awareness, and collection scores from the IUIPC. Interestingly, as well, subcategory scores on the IUIPC also correlate significantly with each other, indicating a connection between measures of control, awareness, and collection.

Our primary goal is in understanding the relationship between attitudes and behaviour. Here, we find significantly weaker correlations. For behaviour, only the collection measure from the IUIPC exhibits a statistically significant correlations, and the overall correlation is quite weak ($r = 0.234 \rightarrow r^2 = 0.055$) meaning that collection score explains only 5% of the variation in attitude score. Alongside behaviour, we also examine participants' knowledge; we note that there is a correlation between quiz scores and the overall IUIPC overall score, but this correlation, while significant to the 0.05 level, is also quite weak, explaining, again, just over 5% of the variation in knowledge scores.

### 4.0.3 Post-Module Results

The online learning module was meant to inform participants about security and privacy risks they might be unaware of with respect to contextual privacy. We envision it as treatment in the presented study with a goal of assessing whether it fosters increased knowledge and ability to align behaviours and attitudes.

During data collection for this study, a subset of answers to the open-ended questions were lost. For transparency, we note that our qualitative data for these questions comes from a subset of our participants from study 1, plus participants from study 2 (see below) who also completed the same post-video-module questions in study 2. Given the consistency across salvaged data from study 1 and from participants in study 2, we note the following:

- When participants were asked if the module presented new information, all but one participant answered yes. In particular, participants' noted that the extent of tracking and the weakness of privacy protections, particularly with respect to contextual privacy through data sets acquired over time. The new knowledge was not that these data sets existed, but the extent of the datasets, i.e. that *"almost everything is tracked and our privacy so weak."*

- When asked whether participants intended to change their behaviours or to reflect on their current behaviours, participants were split. Many participants found the tutorial useful and felt better informed. However, a small subset of our participants reacted negatively, feeling overwhelmed, i.e. : *"I feel a bit upset, but I already knew most of it even before."*.

## 4.1 Discussion of Study 1

One thing that is clear from the above study is that participants' attitudes, as measured by Westin's categorisation and the IUIPC, and their behaviours, as measured by our behavioural questionnaire, are inconsistent. For Westin's categorisation, this mismatch between attitude and practices is perhaps not surprising; King, sampling over 900 participants, noted that " [Westin's] categories were not reliably associated with any of three privacy measures." Dupree et al. [6] stated that one primary motivation for their exploration of

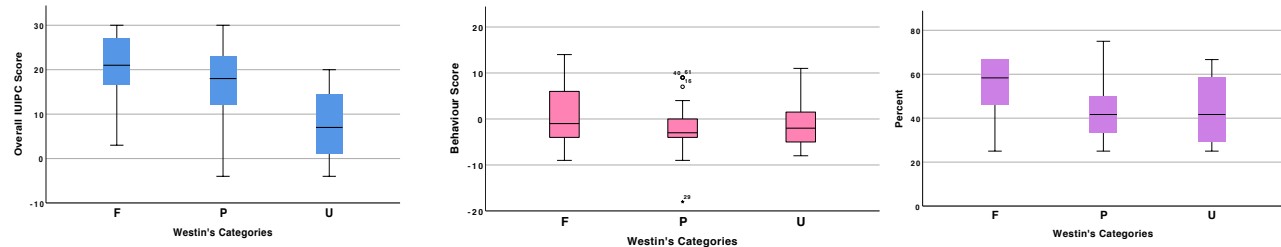

Figure 3: Study I: Attitude, Behaviour, Knowledge/Quiz Score, and Westin's Categories

Table 1: Study I: Pearson's R Correlations for Questionnaires and Quiz Scores

| | | IUIPC | | | | Behaviour-Score | Quiz Score |
| | | Control | Awareness | Collection | Overall | | |
|---|---|---|---|---|---|---|---|
| IUIPC | Control | - | | | | | |
| | Awareness | 0.679** | - | | | | |
| | Collection | 0.469** | 0.600** | - | | | |
| | IUIPC-Overall-Score | 0.770** | 0.826** | 0.864** | - | | |
| Behaviour-Score | | 0.01 | -0.021 | 0.234* | 0.149 | - | |
| Quiz Score | | 0.064 | 0.191 | 0.199 | 0.228* | 0.02 | - |
| *n = 76, * denotes significance at the 0.05 level (two-tailed), ** denotes significance at the 0.01 level (two-tailed)* | | | | | | | |

alternative mechanisms for categorising users was the overall lack of consistency between Westin's categories and user behaviours.

In contrast the IUIPC results were somewhat more surprising. Naresh et al. [19] note that the IUIPC was motivated by psychological principles designed to elicit data the correlates well with behaviours. Their results using the IUIPC show success in predicting intention to disclose behaviours [19], meaning that the IUIPC has been shown to predict behaviours in limited circumstances. In our above correlation analysis, we take the most liberal interpretation of correlation possible, looking only at whether IUIPC measures can explain any of the variance in our behaviour or knowledge scores. Despite this liberal interpretation of relationship, we find only limited and weak correlation between IUIPC and other scores.

We followed this analysis of attitude-behaviour with a concrete intervention, demonstrating the risks to contextual privacy of online activities. Participants were concerned, found the information useful, and expressed a desire to change. These results echo awareness-based results from past work [12], where awareness is linked to desire to change. An open question is whether, with limited information, users actually can effectively modify behaviours.

## 5 STUDY II: INFORMATION RETENTION AND USER ACTION

The goal of our initial study was to explore discrepancy between attitudes and behaviours; a subsequent question is how we might align attitudes and behaviours. Earlier work [3, 12, 27] has presented mixed results regarding users' ability to effectively adjust their behaviour.

In this section, we explore the issue of behaviour change. The goal is primarily to perform a qualitative study exploring whether participants felt they were able to change their behaviours, and what changes they make or aim to make. To assess this, we repeat the study design from study 1, but expand the qualitative component at the end of the study and add a follow-up study where we revisit participants after a two-week break.

Overall, the primary goal of this study is to probe the following research questions: Do participants intend to change their behaviours? Do they follow through on that intent? If so, what do they effectively change and, if not, what inhibits their ability to change? We examine

specifically what changes participants made (in their own words), if any, and why they did or did not change. As a result, we analyse a smaller set of participants, seeking, primarily, thematic results from open-ended question data provided by end users.

### 5.1 Study Design

To preserve consistency, our study replicates the study methodology of Study 1 with three changes. First, we do not consider Westin's categories in this study, as they seem unrevealing based upon the results of study 1. Second, alongside the original questionnaires, we add three additional open-ended questions post-video to the end of the study to more directly gauge user intent. Third, alongside our initial data collection, participants were invited to participate in a follow-up study.

#### 5.1.1 Detailed Protocol

The study was run as a two part longitudinal within-participants study, as follows:

- *Part 1:* Our initial contact with participants follows the methodology outlined in the first study of the paper. Participants were given the demographic questionnaire, the IUIPC [20], the same behaviour questionnaire and the same pre-knowledge quiz from the earlier experiment. We then asked participants to view the online video presented in Study I, complete the module, answer the post-module questions from study 1, and added the three additional questions on intent to change (see directly below and Appendix).

- *Part II:* Two-weeks later, we follow-up with participants. We administer an open-ended survey to understand what they remembered about the video. We also ask if the participant underwent any events, conversations, or changes in the two week period following the the online tutorial. Finally, we asked participants to repeat the same knowledge quiz and also remeasure their IUIPC attitudes.

The questions added to the end of the first part of the study are as follows:

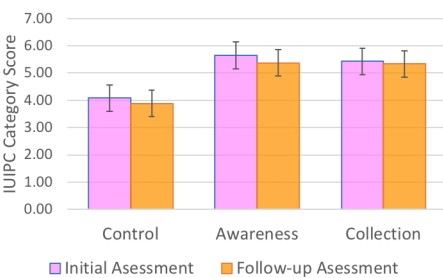

Figure 4: Study II: IUIPC Components and Westin's Categories

1. Do you intend to talk to your friends, family, or co-workers about security and privacy on social media?

2. In your opinion, is the use of the consumer information beneficial to the consumer?

3. Do you find it beneficial when your personal information is collected and used to enhance your online experience?

During the follow-up, we ask the following questions:

1. What do you recall about the tutorial video that was presented to you in your last participation session?

2. What changes have you made to your personal security practices? Why?

3. What changes did you plan to implement and haven't?Why did you plan to change those practices? Why haven't you?

4. Can you describe any social interactions that were affected by the completion of the video tutorial? If so, how did they make you feel?

5. Has there been any other consequences of the video tutorial?

All questionnaires and scoring are included in the Appendix. Table 4 in the Appendix illustrates the questionnaires and study information.

### 5.1.2    Participants

21 participants were recruited for part one of this study, and 9 participants returned for part two of the study. To support follow-up with participants, our data collection procedures were moved to Mechanical Turk under the advisement of the Office of Research Ethics at our institution. We host our survey through Mechanical Turk and Survey Monkey and use Mechanical Turk to pay participants. We require part 1 of our study as a pre-condition on workers for part 2 of our study. We offered $2 for the first module (questionnaires, tutorial, and post survey), and $3 for the follow-up module two weeks later.

### 5.2    Study II: Results

Table 2 contrasts participants from each cohort in our study. The most significant difference in our study was the Behaviour score from the two cohorts. We note that compared to our first study cohort, our second study displays higher scores on the Behaviour questionnaire, i.e. exhibits more positive behavioural scores. However, to note: if one examines Figure 3, the possible scores on the behaviour quiz range from [-24, 24]; given the high standard deviations in Figure 3, it should be unsurprising that these differences are not statistically significant. Knowledge scores and overall IUIPC scores were very similar between the two studies, again not statistically significantly different.

### 5.2.1    Quantitative Data

Because of the structure of the Mechanical Turk + Survey Monkey survey, participants could skip the quiz in whole or in part. Of the 21 participants in the study, 6 chose to omit all or part of the quiz, and these are removed from analysis. In the follow-up, 9 participants responded and 2 choose to skip the quiz. Overall, we see no large change in knowledge as recorded from quiz scores before the video and two weeks after the video. Participants seemed stable at their knowledge level prior to the quiz.

Alongside comparison of mean quiz scores, we also performed comparison of IUIPC scores during both part 1 and part 2 of the study to see if IUIPC scores had changed. Pairwise comparisons found no significant changes in the IUIPC categories or overall IUIPC score. Furthermore, participants self-reported behaviours were not significantly different ($p > 0.05$).

### 5.2.2    Qualitative Analysis Intent and Follow-up

Initially after the intervention, the majority of participants stated that they would indeed need to review their online behaviours, and settings. *"I do my practices because I feel like those are the bare minimum that everyone should be doing. I feel like they aren't enough anymore and I may need to add more"*

However, after two weeks, participants had limited memory of the video explaining issues around contextual privacy. The primary take away for participants specifically focused on the topic of 'social media'.

Alongside limited memory of the details of the video, few participants reported changing any behaviours. Those that did were either of limited use for protecting contextual privacy, or it was ambiguous to determine what specific actions they had altered apart from general awareness:

*"I have changed my password on different websites. It has made me feel a bit more secure."*

*"I feel that I maybe more secure now in my online practices. I take more precautions when I'm online and am more aware of the dangers and risks as well."*

*"Reading the terms and conditions instead of just skimming and accepting them."*

Some participants, while they had not yet changed their behaviour, expressed a desire to modify their behaviour in the future. Examples included limiting on-line time, limiting on-line purchases, or limiting use of social media.

*"I will need to make a list of what is most important to me and then see what I can do to limit my online time."*

*"I am going to try to stop using Gmail but it's connected with so many online things now sadly. I will also not use many apps, buy as many things online, etc."*

*"I have been easing away from Facebook, the only social media I use, for several months. Will probably continue that trend. I may un-friend all my people friends and use it strictly for groups, which I do enjoy."*

However, many participants also expressed exasperation. Knowing that the mismatch exists, sentiments of confusion and frustration coloured their plans of action. Overall, participants reported being unsure how to better match their expectations. *"I'm not sure how I can change what I do that would eliminate some of that besides turning some things off on Google, which i did a few months back."*

*"I don't know what I should do. Google, Verizon, stores, will always know about me"*

## 6    DISCUSSION

Our study contributes a better understanding of the relationship between understanding, attitudes, behaviours and knowledge related to online privacy. The data suggests that people do not completely

Table 2: Summary of IUIPC, Behaviours and Knowledge for All Participants. Note that Study 2 cohort 2B is a subset of cohort 2A.

| Study | Cohort | n | IUIPC | | | IUIPC Overall Score | Behaviour Overall Score | Knowledge Quiz Score |
|---|---|---|---|---|---|---|---|---|
| | | | Control | Awareness | Collection | | | |
| 1 | 1 | 76 | 5.2 | 5.9 | 5.6 | 15.7 | -1.3 | 47.0 |
| 2 | 2A | 21 | 4.1 | 5.7 | 5.4 | 16.9 | 4.4 | 48.4 |
| | 2B | 9 | 3.9 | 5.5 | 5.4 | 15.2 | 2.4 | 47.2 |
| | ALL | 87 | 4.9 | 5.8 | 5.6 | 15.9 | 0.2 | 47.3 |

Table 3: Study III: Follow-up Quiz Performance

| | Participants | Quiz Score |
|---|---|---|
| Initial | 21 | 48.4% |
| Two Week | 9 | 47.2% |

understand security risks: participants reported acquiring new information on online data collection. In analysing attempts to change behaviour, we found limited change (despite a stated desire to change) and some frustration given the challenges associated with change. Since participants stated that their past practices generally were insufficient, we also again observed the privacy paradox, a finding strongly supported by current literature [23], though not with respect to attitudes in contextual privacy. Finally, we found that the IUIPC, a privacy scale that has been shown to correlate well with information sharing behaviours, struggles to correlate with on-line behavioural measures of pro-contextual-privacy behaviours.

## 6.1 Encouraging Behavioural Change

Given data that participants attitudes and behaviours are misaligned, that participants indicate a desire to change, but that participants struggle to find effective ways to address contextual privacy, one question we can ask is how can we aid participants in promoting contextual integrity of their data?

One additional take-away from our user studies is the overall frustration and futility, the learned helplessness of users, once they find out how challenging privacy preservation is in an era of big data and online user profiles. We believe that the design implications from this study advocate for persuading users to change behaviours. Therefore, borrowing from the domain of persuasive technology [7], we briefly highlight avenues for future work clustered around Fogg's design space for persuasive technology: motivation, ability, and triggers. We cluster these design ideas around principles of Unlearning Helplessness and Contextual Awareness.

### 6.1.1 Unlearning Helplessness

Often, participants stated that reducing computer and internet use would be their strategy for privacy protection. However, participants who articulated these goals articulated them as future, or aspirational goals, rather than as concrete actions made in-the-present. Our concern is that these goals may not be feasible and that having no concrete plan of action can thwart motivation [7, 9, 25, 28]. Users may begin to suffer from a learned helplessness [1] if they specify unrealistic goals. Consequently, their in ability saps their motivation.

There do exist an ever-increasing set of ways to enhance privacy. Social networking companies, search providers, and on-line merchants are all providing ways for users to change privacy settings, to opt in or out of data collection, and even to review privacy settings, personal information, and the data collected when services are used. Design options include:

- In the same way that users install tools to encourage them to take a rest break or to encourage them to exercise, user tools could be provided that encourage small privacy awareness check-ups on a daily basis – small mini tutorials, for example, to guide Facebook setting, to install Tor, to browse you Google profile – that would allow users to continually enhance their privacy behaviours and knowledge via simple walk-throughs. This enhances ability and also provides a trigger for positive action.

- Successful practices adopted could be tracked and incentivised (e.g. through awards, badges, or a representation of improvements) such that users continue to take small, concrete steps until their actions align with their behaviours. This provides users with both motivation to continue to improve and an awareness mechanism for how they are currently doing.

### 6.1.2 Contextual Awareness

Revealing information to the 'lay' user is often effective for cautioning for online behaviour attitude change. Swanson, Urner, and Lank used a packet sniffing demonstration to show users just how easy it is to intercept their internet activity [27] and Almuhimedi et al. gave notifications to users every time the user's location data had been collected [3]. The above instances of potential privacy breaches are for in-the-moment privacy breaches: sharing information on an unsecured Wi-Fi network or location access by an application, for example. Beyond the singular moment of use, protecting contextual privacy requires that the user be aware of the inferences that can be made from multiple pieces of information, how information is stored and how information is further traded. However, these same awareness techniques could be adapted to promote contextual privacy:

- The challenge with information sharing is that each new datum collected allows online service providers to develop more enhanced representations of an individual, but there are tools that can prevent this direct attachment of information to an individual. As a simple example, researchers could seek to enhance the usability of privacy enhancing technologies (e.g. Tor, PGP) thus enhancing users ability to protect contextual privacy.

- Researchers could seek to give users a better understanding of the data set that exists on them and when additions occur. Online data comes from the user and, if services can capture and archive it, so, too, can tools on the user's machine. If this data is analysed it can be contrasted during leakage, thus highlighting potential additions to online data before the user acts. This can increase motivation and ability to limit leakage, and can be used to trigger the user toward positive behaviour by eliminating online actions (e.g. browse via the Tor browser, encrypt email message).

Even without tools to act, giving the user the tools to know what exists can be valuable. One challenge that user's have, today, is a lack of awareness of how their privacy is compromised due to online accumulated data in everyday interactions. Continued highlighting of this may frustrate users, but, in some sense, that frustration comes from the fact that they are already compromised. Overall, the question then becomes, "Is ignorance truly bliss?". We posit that awareness – even when action is challenging – can be an important goal of privacy preserving tools.

# 7 CONCLUSION

In this paper, we probe users attitudes, behaviours, and knowledge with respect to contextual privacy. Through two studies, one focusing on behaviours, and a second focusing on the challenges of change, we highlight the challenges user's face in aligning attitudes and behaviours with respect to their desire to protect the contextual integrity of their online data. Given the limited devoted to understanding contextual privacy, we highlight potential tools that can serve to enhance users contextual privacy through an analysis of information leakage, a guidance toward positive practices, and an awareness of the information that currently exists.

## ACKNOWLEDGMENTS

The authors wish to thank A, B, C. This work was supported in part by a grant from XYZ.

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

# APPENDIX: Measures & Materials

This appendix outlines all the measures and materials used in the study. Calculation protocol for each questionnaire is included.

## A WESTIN'S PRIVACY SEGMENTATION

The following questions are from Westin's Privacy Segmentations [14, 29, 30]. The following questions were presented with the instructions: "*Choose what best describes you.*". Participants chose from a five point Likert Scale from *Strongly disagree*, to *Strongly agree*. Neutral was presented as *Do not know*.

1. Consumers have lost all control over how personal information is collected and used by companies.

2. Most businesses handle the personal information they collect about consumers in a proper and confidential way.

3. Existing laws and organizational practices provide a reasonable level of protection for consumer privacy today.

### A.1 Calculation

The calculation of Westin's privacy scores for the presented sections were simple. Answers were scored based on [14]. If participants agreed or rated it as important to them to Q1, and disagreed or rated unimportant to them both Q2 & Q3, they were classified Privacy Fundamentalists. Privacy Unconcerned disagreed on Q1 and agreed to Q2 & Q3. Everyone else was classified as Pragmatist [11].

## B INTERNET USERS' INFORMATION PRIVACY CONCERNS (IUIPC)

The IUIPC questions [20] were presented with the instructions: "*Choose what best describes you.*". Participants chose from a seven point Likert Scale from *Strongly disagree*, to *Strongly agree*. Neutral was presented as *Neither agree or disagree*.

1. Consumer online privacy is really a matter of consumers' right to exercise control and autonomy over decisions about how their information is collected, used and shared.

2. Consumer control of personal information lies at the heart of consumer privacy.

3. I believe that online privacy is invaded when control is lost or unwillingly reduced as a result of a marketing transaction.

4. Companies seeking information online should disclose the way the data are collected, processed and used.

5. A good consumer online privacy policy should have a clear and conspicuous disclosure.

6. It is very important to me that I am aware and knowledgeable about how my personal information will be used.

7. It usually bothers me when online companies ask me for personal information.

8. When online companies ask me for personal information, I sometimes think twice before providing it.

9. It bothers me to give personal information to so many online companies.

10. I am concerned that online companies are collecting too much personal information about me.

To prevent participant fatigue and attrition, we choose to only include directly relevant sections. Therefore, we omitted the following portions of the IUIPC: Omitting Errors, Scenarios, Case Studies, Trusting Beliefs, Risk Beliefs, and Intention to Give Information.

### B.1 Calculation

To score, each question was given a numerical score based on the answer provided by participants. For pro-privacy attitudes, Strongly agree/Very important to me = +2, Neutral/Do not know/I do not know or care = 0, Strongly disagree/Not important to me = -2. Questions which were negative toward privacy had similar but opposite scoring. The sum of these numbers were used as the foundation of the overall *IUIPC* Overall Score. To get the individual category scores of *Control* (items: 1,2,3), *Awareness* (items 4, 5, 6), and *Collection* (items: 7, 8, 9, 10) items were divided into their classified categories according to the IUIPC guidelines and given a numerical value from 1-7 (e.g. Strongly Agree = 7), again reverse questions (meant to test user attentiveness) were given the opposite scores. Blank or no answer was calculated as a 0/Neutral.

## C PRIVACY PRACTICES AND BEHAVIOUR QUESTIONNAIRE

The following questions were presented with the instructions: "*Choose what best describes you.*". Participants then choose from a five point Likert scale ranging from "Does not describe me at all", "Somewhat does not describe me", "Neutral", "Somewhat describes me", and "Completely describes me".

1. Ad Blockers are installed on my devices.

2. I use privacy aware browsers for online searches (for example, Duckduckgo, Tor, etc.)

3. I will download free applications, in trade for some personal information.

4. I disrupt tracking or send do not track requests while I browse online.

5. Google search results are targeted to my location.

6. I check into places or events and/or geotag my posts on social media.

7. I allow third party applications to access my information on social media.

8. I always read terms and conditions on applications or online before hitting agree.

9. I ask not to be checked into places or events or tagged in posts on social media.

10. I list personal identifiable features on social media (example, my causes, sexuality, political views).

11. I review products or stores online.

12. One or more of my social media accounts is public.

### C.1 Calculation

The questionnaire was developed for the study to better understand participant behaviours online. To score, the behaviour scale was rated into negative, neutral, and positive privacy behaviours. Positive privacy behaviours were classified as behaviours taken to protect one's privacy online, the opposite was true of negative behaviours. If a participant responded positively to a behaviour classified as a positive privacy behaviour, they were awarded a positive score (i.e., 2 for "Completely Describes me" or "Very important to me") and the opposite was true for an agreement to a negative privacy behaviour. The average of these numbers were used as the foundation of the overall *Behaviour* Score.

Table 4: Protocol Components of each Study

| Study | n | Demographics | Westin | IUIPC | Behaviour | Quiz | Video | Post-Module (PM) | Open Follow-up (F) |
|-------|---|--------------|--------|-------|-----------|------|-------|------------------|--------------------|
| 1 | 76 | x | x | x | x | x | x | x | |
| 2 (I) | 21 | x | | x | x | x | x | x | |
| 2 (II) | 9 | x | | x | x | x | | | x |

## D  Test-Your-Knowledge QUIZ

The following quiz was presented. Two types of questions were presented: multiple choice and True/False Questions.

### D.0.1  True or False

Participants were given the following questions and asked to identify which they believed to be true or false.

1. Companies can target ads and coupons based on my location. - T

2. Shortened URLs can trick users into visiting harmful sites where personal information can be compromised because the full URL is not seen - T

3. Companies cannot target ads based on previous websites I have visited. - T

4. As long as you have a firewall and antivirus software installed, there is no threat to security - F

5. Phishing schemes and Trojan horses are similar because they both rely on fooling the user into believing they are harmless. - T

6. Even though your data is not being sold, companies can still profit from selling space to advertise. - T

7. It can be difficult to opt out from data collection because companies do not need to disclose what information they have collected about you. - T

8. Companies who collect information must tell you what they know about you and how they know it, upon request. - F

### D.0.2  Multiple Choice

Participants were given the following list of questions and asked to choose the one answer they believed was most correct.

1. Malware is:

   (a) Software downloaded for purposes other than it's intended use

   (b) Software that may be unintentionally downloaded as part of a package

   (c) Software that can capture and record data without the users consent

   (d) All of the above *

2. Which of the following does now appear in Facebook's Terms and Conditions?

   (a) We do not give your content or information to advertisers without your consent

   (b) You understand that we may not always identify paid services and communications as such

   (c) Facebook's license to user's content ends upon the deactivation or deletion of a user's account *

   (d) You will not bully, intimidate or harass any user

   (e) You permit a business or other entity to pay us to display your name and/or profile picture with your content or information, without any compensation to you

3. The company 'Acxiom' claims to have:

   (a) Files on 50% of the world's population, with about 2000 pieces of information per consumer.

   (b) Files on 10% of the world's population, with about 1 piece of information per consumer.

   (c) Files on 1% of the world's population, with about 1500 pieces of information per consumer.

   (d) Files on 100% of the world's population, with about 2 pieces of information per consumer. *

4. Opting out of information collect is:

   (a) Always Free

   (b) Always Easy

   (c) A legal right

   (d) All of the above *

   (e) None of the above

5. What is a data (or information) broker:

   (a) A company or individual that profits from profiling users based on demographic information

   (b) A company or individual that profits from protecting your data

   (c) A company or individual that profits from collect *

   (d) A company or individual that makes privacy and security laws or programs

6. Which of the following threats does not rely on sensitive information:

   (a) Ransomware *

   (b) Spyware

   (c) Key listeners

   (d) All applications profit from collecting sensitive data

*Answers to the quiz above are denoted by an *.

### D.1  Calculation

The calculation for the quiz relied on the answer key. Participants were rewarded one point for every correct answer. The percentage was calculated based on the possible points. 'Blank/No answer' was recorded as incorrect.

## E   Open-ended Questionnaire

The open-ended post-study questionnaire containing questions about how participants' initial security and privacy practices beliefs are affected by the potentially new information introduced in the module. At this point we seek to gain a better understanding of how the participant feels about the current agreements that have been made or legal documentation agreed to with respect to online companies.

- Did the online tutorial teach you new information?

- How do you feel about the information you have read?

- After completing the tutorial, please reflect on your past online security and privacy practices. Do you feel that they are sufficient?

- Do you intend to make any changes to your online security and privacy practices?

### E.1   Analysis

Analysis of the qualitative data was open coded and to understand user's self-reflection on the new information and any intentions they may have moving forward [10, 26].

## F   Two-week Follow-up

The portion of the analysis pertains exclusively to the final study of the paper and is presented at the two week followup.

- What do you recall about the tutorial video that was presented to you in your last participation session?

- What changes have you made to your personal security practices? Why?

- What changes did you plan to implement and haven't? Why did you plan to change those practices? Why haven't you?

- Can you describe any social interactions that resulted from the completion of the video tutorial? If so, how did they make you feel?

- Has there been any other consequences of the video tutorial?

### F.1   Analysis

Similar to the open-coded questions, the data received in this section were also analysed using an open coding protocol [10, 26]. We sought to understand the aftermath of the video and retention of knowledge. Particularly, we wanted to understand the consequences and any impact that the video may have had on their contextual inquiry understanding, and pursuit of any changes.