# OpenReview forum: "Understanding the Contextual Privacy Comprehension, Attitudes, and Behaviours of Users"
_graphicsinterface.org/Graphics_Interface/2023/Conference — Submitted to GI 2023_

### Official Review · Reviewer_hUdN · 2023-01-12
**Great idea, but lack of sample size to conclude any meaningful results**

**Rating:** 5
**Confidence:** 4

**Review:**

This paper investigates the implications of training online users about data collection disclosure and its impact. The authors conducted two studies, the first was to find correlations across characteristics by measuring the general behaviour and knowledge of users, along with education, then the second study was to perform the same study, with an additional follow-up stage with the same participants after two weeks to assess the change. The authors claim the positive results from educating users about understanding contextual integrity and the long-term implications of information disclosure. Finally, the authors provide guidelines for the design of private data-preserving technologies.

In general, this paper is well written especially in describing how the degree of selective disclosure is related to the long-term privacy data leak. Below are my comments:

- The study used a video as part of the education stimulus. I wonder how this behaviour was tracked. Was the completion count also tracked? Were participants able to guess and change until they reach a full 'completion'? What happens when participants (since online) only listened to the video clip, without viewing the visuals?

- How was the score compared to see the impact of education? what makes to say "we see no large change (...)"?

- What method was used to analyze qualitative data? What were the codes? themes? how were the themes validated?
- While qualitative data analysis was useful, it is difficult to conclude only from the limited data size for study II
- I would suggest the article be reframed and claim only from the results of the first study, or reframe the second study result by analyzing deeper into the qualitative data.

---

### Official Review · Reviewer_WgTk · 2023-01-12
**Thoughtfully designed research, with an underdeveloped message**

**Rating:** 6
**Confidence:** 3

**Review:**


This paper explores users’ understanding of contextual integrity and their ability to influence it as they learn through two studies. Overall, I found this to be a reasonably solid though not perfect paper.

I particularly enjoyed the motivation and related work of the paper. I am not an expert in this area (and so I also defer to other reviewers if they identify gaps here that I have not). I found these sections to provide a good introduction to this research space that clearly pulls out the gap the present research is aiming to fill.

This question is examined through two studies. The second study is very similar to the first, but builds on it by examining one of the earlier limitations, by introducing a follow-up session two-weeks following the first to examine how well participants have retained and acted upon the knowledge gained.

I think the studies designed were reasonably robust and well analyzed. A limitation of the work is that it doesn’t do a great job of distinguishing between the impact of the current intervention and the possible impact of a similar (but better designed one). I think the problem with this is that the current intervention (a video) does a pretty nice job of raising awareness of the problem. But it doesn’t provide any guidance on the actions that people can take to improve their privacy or the impact of those actions. We see some of this in the data. E.g., when people discuss ‘giving up gmail’ that seems pretty tied to the fact that gmail was used as the example email system but it remains unclear if people plan to give up email entirely or just switch systems to a potentially equally problematic system. Likewise some of the aspirational actions described seem pretty heavy-handed and unrealistic. Is this because people still lack the knowledge for *how* to improve their privacy.

Again I don’t think this invalidates the data of the study, only that the contribution would be stronger if these issues were better discussed. The conclusion that people can’t act even after they’ve been informed seems a bit brash when the education given only covers raising awareness of the issues and not how to respond to them effectively. As such, I’ve marked this paper as borderline. I’d like to see it move forward because I think it’s good work but I also think that it needs a bit more to really get it firmly over the threshold.

---

### Official Review · Reviewer_FkZ4 · 2023-01-17
**Interesting but flawed study on how users perceptions of contextual provacy alter their ideas of behaviour change.**

**Rating:** 5
**Confidence:** 4

**Review:**

The authors present a study of users' attitudes towards the contextual privacy of their data and whether education changes their plans for how they deal with issues around it.  While the topic is highly relevant and important, and the field is surging and changing  by the minute,   the study contributions are very diffuse and indeed the findings such as they are do not provide much actual pragmatic insight on what we can do about it, short of what is already emerging in the literature and the discussions of privacy, security and data governance overall.
I really wanted to love this paper, but I cannot quite tease out that the essential contributes are.

Then authors carried out two parts to this study: the first looked for whether there was a relationship between attitudes and knowledge around contextual privacy and behaviours between users, and the second examined whether an educational intervention used in the first study produced differences attitudes and behaviours over time.   The novelty of these studies largely lies in the authors' instrument to measure behaviour - a set of 12 questions related to what the participants report they do with respect to privacy behaviours online. do

The lack of a clear statement of what these contributions actually are an be addressed with some re-writing, but there are other issues that unfortunately are related to the design and instruments of the study itself that result in a mismatch between what the study examined, how it did it, and the research questions and subsequent claims the authors make.

Study method, questions and continuity.  In the first study, 76 participants' attitudes and behaviours were examined across three separate models before they completed an online learning tool about privacy. that included a video. The authors then conducted a semi-structured (open-ended) interview to assess the impact of the learning intervention on attitudes, knowledge and awareness.
Study 2 replicated the method of study 1 but added a follow-up study after 2 weeks to see whether ,here was an actual effect of behaviour change after the learning.

The challenge of attitudes and behaviours: The real problem with many of these studies, and this one is no different, is that when you ask people what they are going to, or what they do, you don't actually know if they are doing it. So in the first study, it is not surprising that was no correlation between attitudes and behaviours, because the questions didn't really target - why not? what is about the actions that do not seem to get at the attitudinal concerns? And when trying to measure behaviour change as a reorientation of behaviours over time,   a study needs to observe or collect data in a structured longitudinal schedule. @ weeks, with two observations (pre and post), can measure simple things like memory, but cannot capture the nuances of shifts in what people actually do. Let alone that the authors are depending on only asking people what they do, rather than considering or instrumenting what they actually do.

Thus the results they report are unsurprising to say the least. Behaviour change has been extensively studied with respect to information and awareness over the past decades, notably in the domains and more recently in energy conservation.   The authors would be well advised to review this behaviour literature, it might give them some insights into how to refine these kinds of study designs.  The authors' recommendations and suggestions for better tools to promote awareness, motivation and trigger actions echo well known research in other domains, albeit not necessarily only from the domain of privacy.

A second set of issues relates to the study design itself and the reliance on the online learning module to predict or influence changes in attitudes. One might say that the weakness in the results could have been an artefact of a poor online intervention rather than the larger question of whether information serves to provoke behaviour change. As a data communication research, I am extremely interested in knowing what kinds of interventions might perform differently, not just vaguely whether one type or instance of intervention stands in as a proxy for ANY  such educational approach.   It's a leap to claim that this intervention was sufficiently rich or representative enough to tease out why information did not provoke a more target-goal oriented response in actions.

A related weakness with the second study is that only 9 participants actually completed the last part of the study that collected data about "change". This in addition to the fact that some of the participants had already seen he information in the first study, and there most definitely would be learning effects.

Overll, the work is extremely well motivated, but the study as reported suffers from an over-ambitious goal within a limited scope. A well written limitations section, and a more informed reflection,  might go some way to reducing this problem.   But overall, there si little here that can actually inform designers and researchers about what they can do next that is not already known in the larger context of persuasive technologies, and their frustrating and continuing lack of more impactful success.

---

### Meta-Review · Area_Chair_4Yg6 · 2023-01-18

**Recommendation:** 5
**Confidence:** 3

**Metareview:**

This paper investigates the implications of education for online users about data collection disclosure, the contextual privacy of their data and its impact. The first study was to find correlations across characteristics by measuring the general behaviour and knowledge of users, along with education, then the second study was to perform the same study, with an additional follow-up stage with the same participants after two weeks to assess the change. The authors claim the positive results from educating users about understanding contextual integrity and the long-term implications of information disclosure. Finally, the authors provide guidelines for the design of private data-preserving technologies.

The reviewers consider the topic of the paper interesting and provide interesting motivations and a well-written limitations section.  while study contributions are very dispersed and the findings do not provide much sensible insight into what we can do about it. One novelty of the paper may be the instrument to measure behaviour, which is consisted of 12 questions related to what the participants report they do with respect to privacy behaviours online.

The paper seems to lack a clear statement of what their contributions are, and the study design could have been improved by focusing on measuring the effect of education. The claimed results do not seem to be pragmatic. The research questions must have corresponding study designs and instruments. The data collected was not on par with the consequent claims, thus it can be revised by collecting more data, or re-running the study. The reviewers appear to be concerned about the study design to look at the correlation between attitudes and behaviours as the data collection instruments do not focus on the actual behaviours while the education is only to raise awareness of the issues not how to respond or resolve the issues.

Some of the factors mentioned in the review can be revised by reframing and thus re-writing to focus on a specific novelty and especially contributions. It is advised to look at the literature about studies on behaviour change in the field of energy conservation. Also, the study needs to consider a structured longitudinal design in order to see the behavioural changes over time. Finally, the learning effects should be considered when designing the study.

The reviewers consider the paper slightly below the acceptance threshold and need more revision in order to be accepted.